# SEAM: Semantically Equivalent Across Modalities Benchmark for Vision-Language Models

**Zhenwei Tang[1], Difan Jiao[1], Blair Yang[1,2] & Ashton Anderson[1]**
[1] Department of Computer Science, University of Toronto
[2] Coolwei AI Lab
{josephtang,difanjiao,blair,ashton}@cs.toronto.edu

## Abstract

Evaluating whether vision–language models (VLMs) reason consistently across representations is challenging because modality comparisons are typically confounded by task differences and asymmetric information. We introduce **SEAM**, a benchmark that pairs semantically equivalent inputs across four domains that have existing standardized textual and visual notations. By employing distinct notation systems across modalities, in contrast to OCR-based image-text pairing, SEAM provides a rigorous comparative assessment of the textual-symbolic and visual-spatial reasoning capabilities of VLMs. Across 21 contemporary models, we observe systematic modality imbalance: vision frequently lags language in overall performance, despite the problems containing semantically equivalent information, and cross-modal agreement is relatively low. Our error analysis reveals two main drivers: textual perception failures from tokenization in domain notation and visual perception failures that induce hallucinations. We also show that our results are largely robust to visual transformations. SEAM establishes a controlled, semantically equivalent setting for measuring and improving modality-agnostic reasoning. We publicly release the code, dataset, and leaderboard to foster further research.

## 1 Introduction

Vision-Language Models (VLMs) have made rapid progress in understanding and generating content that spans visual and textual domains, making tangible steps towards more general artificial intelligence (Li et al., 2023; Liu et al., 2023c; OpenAI, 2024a). As these models are deployed more broadly, however, it becomes necessary to measure whether they reason consistently across representations, and assess whether multimodal models have general, integrated understanding or approach problems with narrow, modality-specific processing. The ability to solve tasks across multiple modalities is not, by itself, evidence of unified reasoning, as performance could depend on how information is represented.

A fundamental obstacle with comparing a VLM's ability across modalities is *confounding by modality*: comparisons between vision and language typically vary both the representation and the task, making it unclear whether observed performance differences reflect genuine reasoning gaps or variable task difficulty (Lu et al., 2023; Yue et al., 2024a). Even for a fixed concept, textual and visual instances rarely have matched semantics and difficulty, and asymmetric information content further obscures what is being measured. This lack of standardization across modalities makes it difficult to isolate and measure core multimodal processing capabilities. Existing approaches either lack rigorous cross-modal alignment or introduce biases through asymmetric information content, leaving the field without a principled way to measure modality-agnostic reasoning.

We introduce **SEAM**, short for Semantically Equivalent Across Modalities, a benchmark designed to rigorously assess modality-agnostic reasoning in VLMs by holding semantics constant while varying only representation. **SEAM** leverages domains with standardized notation systems in both language and vision modalities—chess (FEN notation vs. board

images), chemistry (SMILES strings vs. structural diagrams), music (ABC notation vs. sheet music), and graph theory (adjacency matrices vs. node-edge diagrams)—to ensure semantic equivalence: a property where representations preserve identical meaning despite being presented in different modalities. Each task is self-contained within a single modality, eliminating confounding factors from joint inference and enabling clean language-only, vision-only, and language-vision evaluation. The benchmark comprises 16 tasks (four per domain), and 200 items per task (3,200 total), formatted as multiple-choice questions with carefully constructed distractor answers to calibrate difficulty.

Our evaluation of 21 state-of-the-art VLMs reveals systematic modality imbalance: all models exhibit significant gaps between vision and language performance. Additionally, cross-modal answer agreement is relatively low, often not far from a random baseline, suggesting that models differ substantially in how they process information across modalities, and have substantial room to improve in integrating reasoning and leveraging abilities across representations. Furthermore, we observe that modality imbalances vary significantly across domains. Finally, our error analysis highlights two recurring failure modes: (i) textual perception failures in tokenizing strings in textual inputs (e.g., SMILES, FEN), and (ii) visual perception failures that induce hallucinations. We also perform robustness checks that show our results are not sensitive to common visual transformations.

Our contributions are threefold. First, we introduce **SEAM**, the first benchmark to systematically control for semantic equivalence across modalities, enabling fair evaluation of cross-modal reasoning. Second, we conduct a comprehensive empirical study across 21 models, measuring controlled cross-modal imbalances for the first time. Third, we analyze errors and discrepancies across tasks and models, and pinpoint perception-driven failure modes in modern VLMs that lower agreement rate across modalities, providing actionable insights for future research. **SEAM** provides a principled framework for measuring progress toward more robust and genuinely intelligent VLMs.

## 2  Related Work

**Vision-language models.** Early large VLMs separately process visual and textual inputs with two-stream architectures (Lu et al., 2019; Tan & Bansal, 2019; Chen et al., 2020), followed by unified models for both understanding and generation (Zhou et al., 2020; Zhang et al., 2021; Li et al., 2020). The Flamingo and BLIP families of models (Alayrac et al., 2022; Awadalla et al., 2023; Li et al., 2022; 2023; Dai et al., 2023) shift to bridging pre-trained vision and language models. The integration of LLMs into VLMs has led to powerful improvements such as the MiniGPT family (Zhu et al., 2023; Chen et al., 2023), LLaMA-Adapter family (Zhang et al., 2023b; Gao et al., 2023), and the LLaVA family (Liu et al., 2023c;b; 2024a; Li et al., 2024c;a), which have progressively enhanced visual instruction-following capabilities. Among proprietary models, the GPT series (OpenAI, 2023; 2024a;b; 2025a; 2024c; 2025c;d;b), the Claude series (Anthropic, 2024; 2025a;b), and the Gemini series (DeepMind, 2023; 2024a; 2025a; Comanici et al., 2025) have demonstrated state-of-the-art multimodal reasoning. Strong open-source alternatives have emerged including notable models such as the LLaMA series (Touvron et al., 2023a;b; Grattafiori et al., 2024), the Gemma series (DeepMind, 2024b;c; 2025b), the InternVL family (Chen et al., 2024c;b;a; Zhu et al., 2025), the Qwen family (Bai et al., 2023; Yang et al., 2024a;b; Wang et al., 2024b; Bai et al., 2025; Xu et al., 2025), and Pixtral (Agrawal et al., 2024).

**VLM benchmarks.** Early benchmarks such as VQA (Antol et al., 2015; Goyal et al., 2017), OK-VQA (Marino et al., 2019), and MSCOCO (Lin et al., 2014) were instrumental in evaluating basic visual understanding. Recent benchmarks have expanded the scope to cover more complex capabilities (Yin et al., 2023; Xu et al., 2024; Li et al., 2024b; Liu et al., 2024b; Tong et al., 2024; Yu et al., 2023; Jiang et al., 2024; Ying et al., 2024; Fu et al., 2024), including hallucination detection (Cui et al., 2023; Liu et al., 2023a) and advanced reasoning (Lu et al., 2023; Zhang et al., 2024a). The MMMU series (Yue et al., 2024a;b) introduces college-level multimodal questions and addresses shortcut issues by enforcing vision-dependent evaluation. EMMA (Hao et al., 2025) further emphasizes complex, multi-step reasoning

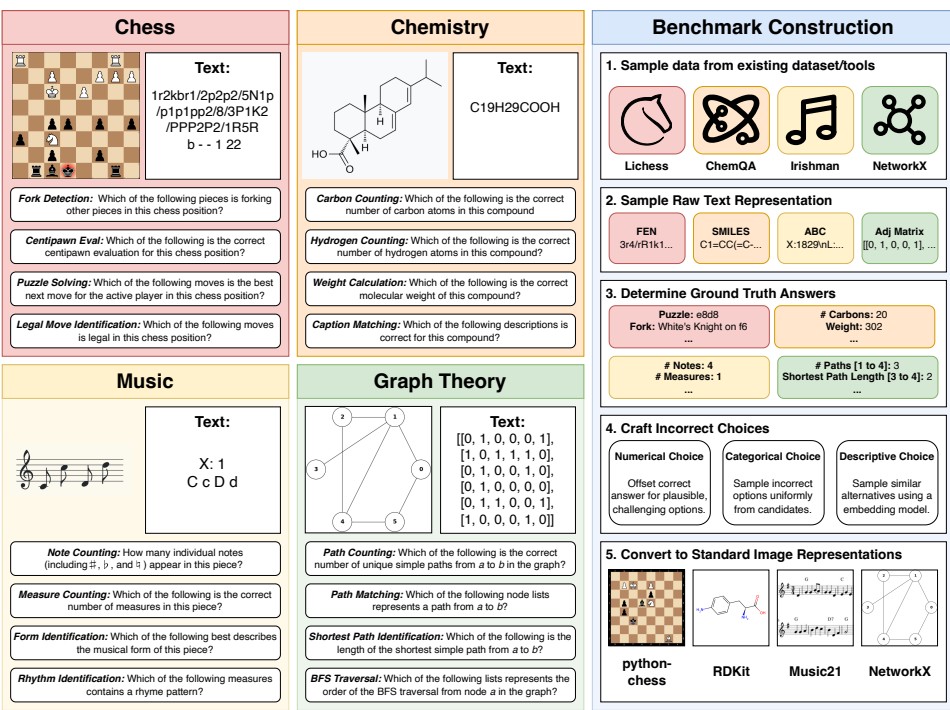

Figure 1: **SEAM** includes 16 tasks in chess, chemistry, music, and graph theory domains with paired visual-spatial and textual-symbolic representations that are semantically equivalent.

across STEM fields. However, assessing the modality imbalance of VLMs remains difficult due to the lack of benchmarks with semantically equivalent inputs across modalities. While previous efforts have attempted to benchmark modality imbalance, they exhibit significant limitations. Yue et al. (2024b) and Zhang et al. (2024b) introduced OCR-derived semantically equivalent image-text pairs, e.g., screenshots or photos of textual questions, but these merely transform text into images without leveraging *distinct notation systems* in different modalities. Thus, OCR-based benchmarks mainly evaluate symbol recognition rather than modality-agnostic reasoning capabilities. A robust evaluation should contrast visual-spatial representations (encoding through spatial relationships and visual patterns) with textual-symbolic representations (encoding through abstract symbols and formal notation) to effectively assess modality imbalance. Further discussions on modality imbalance and domain-specific VLMs are in Appendix A.

## 3 SEAM: Semantically Equivalent Across Modalities Benchmark

In this section, we describe how we designed and instantiated the SEAM benchmark. We begin with our design principles that ensure semantic equivalence across modalities: standardized notations, faithful conversions, and single-modality self-containment. Guided by these principles, we first select four domains that satisfy them (chess, chemistry, music, and graph theory) and then define tasks within each domain, carefully constructing questions for each task. The result is a 16-task, 3,200-item benchmark for attributing performance differences to input modality rather than task confounds.

### 3.1 Design Principles

**Standardized notation in both modalities.** We selected benchmark domains that have well-established, standardized notation systems that can represent semantically equivalent

information across both visual and textual modalities. In chess, positions can be precisely encoded as visual 2-D chessboards or as textual Forsyth-Edwards Notation (FEN) (Edwards, 1994) strings, with perfect semantic equivalence between these representations. Chemical compounds offer a similar duality through structural diagrams that visually represent molecular bonds and atoms versus SMILES (Simplified Molecular Input Line Entry System) (Weininger, 1988) strings that encode the same information textually. Musical compositions can be expressed through traditional visual staff notation with notes and measures or through ABC notation (Walshaw, 2004), which captures musical elements in plain text. Graph theory provides another ideal domain with node-edge diagrams for visual representation versus adjacency lists or matrices for textual encoding. This bidirectional mapping property ensures that information content remains consistent across modalities, enabling rigorous evaluation of cross-modal reasoning. While these notations provide strong semantic alignment, we acknowledge that achieving absolute equivalence impervious to subtle, low-level visual rendering variations (e.g., specific line styles, exact node positioning) is a theoretical ideal; our focus is on ensuring that the essential symbolic information is rigorously preserved across modalities. We later verify that our results are robust to rendering changes.

**Tool availability and real-world prevalence.** Our domain selection is also driven by the availability of robust tools that support faithful conversion between standardized textual and visual representations, a prerequisite for creating semantically equivalent cross-modal pairs. The python-chess library (Fiekas, 2025) provides conversion between FEN notation and visual chessboards; RDKit's capabilities (RDKit Development Team, 2025) allows transformation between SMILES strings and molecular structure diagrams; Music21 (Cuthbert & Ariza, 2010) converts between ABC notation and standard staff notation images; and the NetworkX (Hagberg et al., 2008) library handles both adjacency matrices and node–edge diagrams. These tools not only support cross-modal conversions but also enable automated generation of synthetic questions and answer options across modalities.

We also prioritize domains with widespread real-world use in both visual and textual forms. Chess practitioners routinely alternate between board visualizations and FEN notation on platforms such as Chess.com (Chess.com, 2025) and Lichess (Lichess Team, 2025b). In chemistry, researchers alternate between structural diagrams and SMILES strings, with databases (Kim et al., 2025) providing both. Music analysis involves translation between staff notation and formats like ABC notation (Roland, 2002), and graph theory researchers employ both visualizations and adjacency matrices (Leskovec & Sosič, 2016). This dual-modality prevalence ensures that SEAM evaluates reasoning capabilities that directly transfer to real-world applications.

**Self-contained in each modality.** A key criterion is that every problem should be fully solvable using only its textual representation or only its visual representation. By crafting tasks so that all necessary information is self-contained within each modality, we remove any need for joint inference or reliance on secondary cues. This approach enforces a strict evaluation of cross-modal reasoning and precludes models from exploiting superficial patterns that might appear exclusively in one modality. Consequently, performance in our benchmark reflects a model's true ability to handle semantically equivalent information across different representations, mirroring real-world scenarios where experts freely alternate between visual and textual formats without losing crucial information.

### 3.2 Benchmark Construction

Guided by these principles, we construct 16 tasks (four per domain), formatted as 4-way multiple-choice questions, following MMMU (Yue et al., 2024a). Each task contains 200 items (3,200 total). Detailed construction procedures and hyperparameters for each task appear in Appendix C.

**Text representations and ground truth.** We first sample raw textual representations from datasets or generate them with domain-specific tools. For chess, we extract FEN strings from the Lichess Open Database Puzzles (Lichess Team, 2025c) and Evals (Lichess Team,

2025a). For chemistry, we use SMILES strings from ChemBench (Guo et al., 2023) and ChemQA (Zimmermann et al., 2024; Zhu et al., 2024). For music, we collect ABC notation from the Irishman dataset (Wu et al., 2023) and the Music Theory dataset (Seeker38, 2025). For graphs, we generate adjacency matrices with NetworkX.

Some ground-truth labels cannot be extracted automatically. For example, free-text captions for chemical compounds require human annotation, so we sample them from existing datasets. By contrast, when ground truth is deterministically derivable from the textual representation, such as breadth-first search traversal order in graphs, we generate it automatically with domain-specific tools.

**Calibrated task difficulty.** We calibrate task difficulty when constructing distractors (incorrect options) to enable meaningful cross-modal comparisons. Tasks that are too easy result in near-perfect performance across all modalities, obscuring any potential gaps, while overly difficult tasks reduce performance to random guessing levels, also making it difficult to distinguish between modalities. Maintaining moderate difficulty exposes genuine modality imbalances that can be properly measured. **SEAM** includes three option types—numerical, categorical, and descriptive—each with tailored strategies to produce plausible, appropriately challenging distractors. For numerical tasks, we perturb the correct answer by task-specific amounts (e.g., $\pm300$ centipawns for chess evaluations) to generate alternatives that remain within a plausible range while avoiding near-duplicates. For categorical tasks (e.g., selecting music forms or legal chess moves), distractors are sampled uniformly from the label set, excluding the correct label. For descriptive tasks involving rich text (e.g., chemical compound captions), we use the Multilingual-E5-large-instruct model (Wang et al., 2024a) to retrieve semantically similar alternatives, as random negatives are typically insufficiently confounding.

**Text to image conversion.** For domains with highly asymmetric layouts, such as music staff notation, we pad the images with white space to produce square images. This preserves visual clarity and ensures that structural patterns remain recognizable to VLMs even after automatic resizing. All images are rendered at a standardized resolution of 400×400 pixels to balance detail and computational efficiency. We adopt the default settings of each conversion tool described in Section 3.1 to preserve semantic equivalence with the original text representations, ensuring that neither modality introduces additional information. Importantly, instead of rendering text as images (e.g., via screenshots), we use distinct, standardized notation systems for language and vision modalities. This design targets VLMs' ability to understand and reason over semantically equivalent content across modalities, rather than relying on OCR-style recognition.

Since question–answer pairs are generated with domain-specific tools or sampled from large-scale datasets (e.g., the Lichess Puzzle dataset with over 4.8 million entries), this design enables continual expansion and regular updates, reducing the risk of benchmark leakage from pretraining exposure (data contamination). As models improve, our distractor-generation procedure supports flexible retuning of task difficulty, enabling the creation of more challenging benchmark versions in future releases.

## 4 Experiments

### 4.1 Experimental Settings

We evaluate a total of 21 state-of-the-art vision-language models (VLMs) under zero-shot chain-of-thought prompting (Kojima et al., 2022), using the latest publicly available release of each model series at the time of evaluation. We use the vLLM framework (Kwon et al., 2023) on 8 A100 GPUs to run inference for open-source models, adopting each model's default prompt format, system prompts, and generation hyperparameters when applicable, as detailed in Appendix C We follow the OpenCompass (Contributors, 2023) protocol to extract final answers from model outputs using an external LLM when rule-based extraction fails. As final answer extraction is relatively straightforward, we use the smaller and open-source Qwen2.5-7B-Instruct model for this task, given its strong performance and efficiency.

Table 1: Leaderboard of proprietary and open-source VLMs across language (L), vision (V), and vision-language (VL) modalities. Models are sorted by agreement between language and vision modalities. Bold and underlined values indicate best and second-best performance within each category, respectively.

| Model | Accuracy | | | | Agreement | | | |
|---|---|---|---|---|---|---|---|---|
| | L | V | VL | Avg | L - V ↓ | L - VL | V - VL | All |
| *Proprietary Models* | | | | | | | | |
| GPT-5-mini | 0.787 | **0.653** | 0.830 | 0.756 | **0.630** | 0.846 | 0.653 | 0.584 |
| GPT-5 | 0.804 | 0.632 | **0.857** | **0.765** | 0.627 | **0.876** | **0.657** | **0.596** |
| Claude-3.7-Sonnet | 0.743 | 0.591 | 0.679 | 0.671 | 0.594 | 0.715 | 0.624 | 0.506 |
| Claude-4.1-Opus | **0.827** | 0.578 | 0.814 | 0.740 | 0.575 | 0.844 | 0.580 | 0.523 |
| Claude-4-Sonnet | 0.808 | 0.545 | 0.803 | 0.719 | 0.569 | 0.834 | 0.566 | 0.508 |
| Claude-3.5-Sonnet | 0.665 | 0.560 | 0.514 | 0.580 | 0.537 | 0.549 | 0.508 | 0.378 |
| GPT-4o | 0.635 | 0.482 | 0.627 | 0.581 | 0.503 | 0.686 | 0.532 | 0.410 |
| GPT-5-nano | 0.699 | 0.510 | 0.753 | 0.654 | 0.500 | 0.771 | 0.516 | 0.432 |
| GPT-4o-mini | 0.555 | 0.411 | 0.529 | 0.498 | 0.480 | 0.650 | 0.518 | 0.379 |
| Claude-3.5-Haiku | 0.530 | 0.433 | 0.496 | 0.486 | 0.479 | 0.556 | 0.534 | 0.346 |
| *Open-Source Models* | | | | | | | | |
| Qwen2.5-VL-72B-Instruct | **0.547** | **0.475** | **0.519** | **0.514** | **0.447** | **0.504** | 0.532 | 0.318 |
| InternVL3-78B | 0.525 | 0.427 | 0.482 | 0.478 | **0.447** | 0.498 | 0.487 | 0.293 |
| gemma-3-27b-it | 0.516 | 0.428 | 0.450 | 0.465 | **0.447** | 0.497 | **0.575** | **0.325** |
| gemma-3-12b-it | 0.458 | 0.401 | 0.429 | 0.429 | 0.419 | 0.474 | 0.543 | 0.297 |
| InternVL-2.5-78B | 0.448 | 0.414 | 0.459 | 0.440 | 0.415 | 0.485 | 0.523 | 0.309 |
| InternVL3-8B | 0.382 | 0.357 | 0.386 | 0.375 | 0.388 | 0.425 | 0.456 | 0.229 |
| Llama-3.2-90B-Vision-Instruct | 0.434 | 0.384 | 0.439 | 0.419 | 0.384 | 0.460 | 0.443 | 0.253 |
| Qwen2.5-Omni-7B | 0.363 | 0.354 | 0.364 | 0.360 | 0.353 | 0.375 | 0.375 | 0.183 |
| Qwen2.5-VL-7B-Instruct | 0.303 | 0.350 | 0.359 | 0.337 | 0.347 | 0.389 | 0.437 | 0.216 |
| InternVL-2.5-8B | 0.324 | 0.337 | 0.334 | 0.332 | 0.324 | 0.340 | 0.436 | 0.196 |
| Llama-3.2-11B-Vision-Instruct | 0.289 | 0.330 | 0.323 | 0.314 | 0.287 | 0.303 | 0.401 | 0.152 |

As shown in Table 2 in Appendix B, the extraction results are highly consistent with those obtained using different LLMs.

## 4.2 Results

**General performance comparison.** As shown in Table 1, proprietary models significantly outperform open-source alternatives across all modalities in general, with GPT-5 demonstrating superior accuracy (0.765) compared to the highest-performing open-source model, Qwen2.5-VL-72B-Instruct (0.514). Performance consistently follows scaling laws across both proprietary and open-source model families, with larger variants outperforming their smaller counterparts (GPT-5 at 0.756 vs. GPT-5-nano at 0.654; Gemma-3 27B at 0.465 vs. 12B at 0.429), highlighting the impact of model scale on multimodal reasoning capabilities.

**Vision-language modality imbalance.** In addition to accuracy, we calculate the agreement rate between extracted final answers across semantically equivalent questions presented in different modalities. For each sample, the agreement between modalities is binary (either 0 for disagree or 1 for agree), and these binary values are then averaged across all samples to obtain the overall agreement rate.

In principle, an intelligent reasoner, either human or artificial, could (and perhaps should) exhibit consistent performance when presented with semantically equivalent information in different modalities. For example, a molecule shown as a structural diagram or described through a SMILES string contains the same information. A human expert chemist might prefer the visual structural diagram for intuitive reasoning, but when given a SMILES string, they can still identify the compound and its properties, possibly by first mentally or physically sketching the visual structure. The challenge lies in the molecular complexity

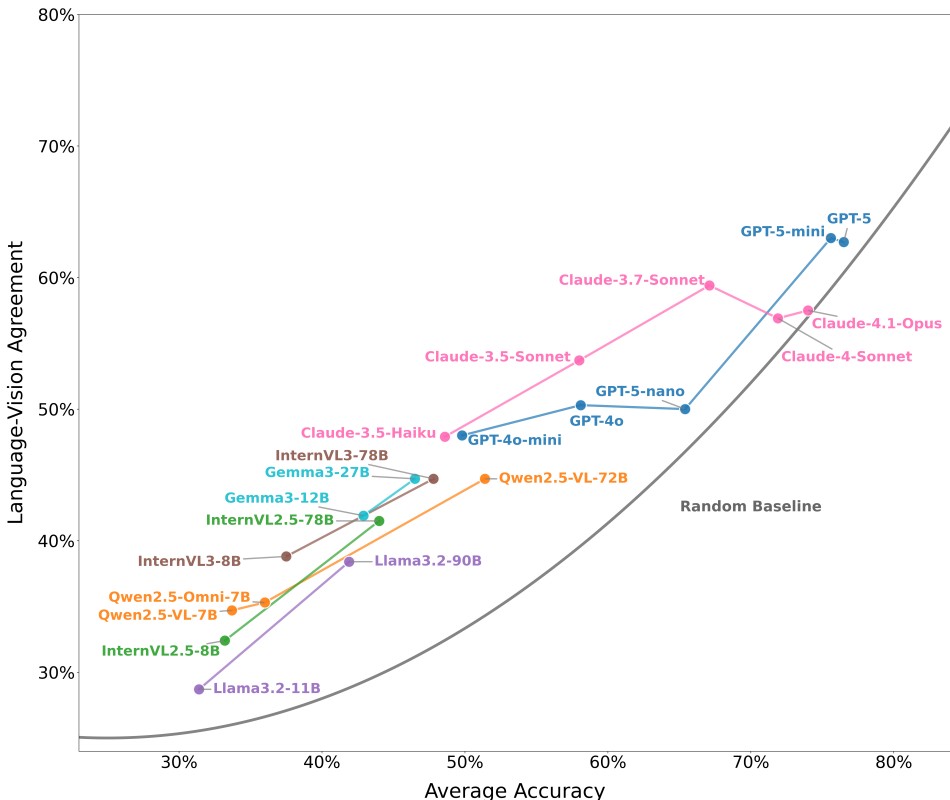

Figure 2: Correlation between final answer agreement rates with language and vision inputs and average accuracy across various VLMs. Models are color-coded by family. Random Baseline denotes how often two models with identical accuracy $p$ would agree by random chance, which can be thought of as a lower bound for cross-modal agreement of real multimodal models.

and properties, not in the modality of its presentation. A truly intelligent VLM could behave analogously, and either naturally treat different input representations similarly, or learn to translate between modalities when it is advantageous to do so—and thus have high agreement between modalities when presented with semantically equivalent information across different modalities.

However, this is not what we observe. The results in Table 1 reveal substantial performance variation (low agreement) across modalities for all evaluated models. Proprietary and larger open-source models consistently achieve higher accuracy with language inputs and lower accuracy with vision inputs. Because VLMs typically build upon extensively pre-trained LLMs, they inherit their reasoning capabilities from a base LLM. Textual inputs are fed directly into the LLM, while visual inputs must first pass through a visual encoder and a projector before being integrated into the LLM, which is a noisy process.

Note that a VLM could theoretically achieve near-perfect agreement across vision and language on our semantically equivalent tasks, even at low accuracy levels. One conceptually simple method would be to learn how to translate between modalities, which is always possible by construction for SEAM tasks, attempting to solve problems in both modalities, and picking one of the solutions. More ideally, a generally intelligent model could learn how to leverage the strengths of each modality, and in which situations it is wise to prefer one modality over another. But none of the VLMs we tested have high agreement, meaning there is substantial progress to be made in attaining fluid and general intelligence that adapts across modalities.

**Correlation between agreement and accuracy.** We observe a strong positive correlation between cross-modal agreement and overall accuracy across most model series in Fig 5, suggesting an association between modality alignment and performance. This raises an important question about the direction of causality: Does higher accuracy lead to higher agreement, or does higher agreement drive better accuracy?

Note that as models become more accurate, they naturally converge as they increasingly agree on correct answers across modalities, artifactually increasing agreement. In the theoretical limit of perfect accuracy, cross-modal agreement would necessarily reach 1 as all modalities output identical correct answers. However, evidence from Zhang et al. (2024b) challenges the notion that accuracy improvements are the *sole* driver of higher agreement. Their findings demonstrate that accuracy and agreement can vary independently within practical performance ranges: VLMs can exhibit high agreement despite poor performance, or achieve high accuracy with relatively low agreement between modalities. This pattern is clearly visible in Fig 5, particularly among models in the GPT and Claude series. This decoupling suggests that the strong positive correlation we observe cannot be explained solely by accuracy driving agreement.

To measure the artifactual relationship between agreement and accuracy, we define a random baseline that represents how often two models with identical accuracy $p$ would agree by random chance, which can be thought of as a lower bound for cross-modal agreement of real multimodal models. We simulate two independent "modalities" with identical accuracy $p$ but no genuine coordination. For a multiple-choice question with 4 options where one is correct, each modality independently selects the correct answer with probability $p$ and each incorrect answer with probability $\frac{1-p}{3}$. The agreement rate between the two modalities is $P = P_1 + P_2 = p^2 + 3 \times \left(\frac{1-p}{3}\right)^2 = p^2 + \frac{(1-p)^2}{3}$, where $P_1$ is the probability both select the correct answer and $P_2$ is the probability both select the same incorrect answer. By varying $p$, we generate the random baseline curve shown in Fig 5.

Comparing this random baseline with our measured results of contemporary multimodal models suggests that current models do not achieve significant cross-modal alignment beyond the simple agreement achieved by task correctness. Most models cluster much closer to this random baseline than to the ideal upper bound of perfect agreement, suggesting that what appears to be improved cross-modal alignment as models scale may simply reflect the mathematical constraint that higher accuracy necessitates higher agreement rates, rather than representing true progress toward unified multimodal understanding.

**Domain-specific modality imbalance.** As illustrated in Fig 3, the degree of modality imbalance is highly domain-specific. In chess and chemistry, models frequently demonstrate comparable or slightly superior performance with vision inputs compared with language inputs. However, this pattern reverses in music, where language inputs generally yield better results than vision inputs. This imbalance becomes even more significant in graph-related tasks, where the performance gap between language and vision inputs substantially widens. These domain-specific asymmetries in input modalities suggest that cross-modal consistency varies significantly depending on the reasoning domain. Such observations motivate us to investigate the underlying mechanisms behind domain-specific performance disparities.

**Textual perception error.** We examine the tokenization processes of open-source models to understand how they parse domain-specific text inputs. In chemistry tasks, we find that SMILES strings such as *COC(=O)C(OC(C)(C)C)c1cc([N+](=O)[O-])ccc1-c1ccc2c(c1)CCCO2* are often incorrectly segmented into semantically meaningless subwords like "**OC**", "**cc**", and "**([**". Such tokenization errors are particularly problematic, where structural elements like parentheses (indicating branches) and square brackets (denoting charged atoms) carry precise molecular semantics that, when incorrectly parsed, lead to fundamentally different chemical interpretations. Similarly, we also identified severe tokenization errors in chess FEN notation examples. However, music ABC notation exhibits fewer issues, likely due to its abundant punctuation markers that provide clearer tokenization boundaries. Interestingly, graph-related tasks, where adjacency matrices primarily consist of commas, 0s, and 1s,

**Chess / Chemistry**

| Model | Chess Language | Chess Vision | Chess Vision Language | Chemistry Language | Chemistry Vision | Chemistry Vision Language |
|---|---|---|---|---|---|---|
| GPT-5 | 0.710 | 0.746 | 0.734 | 0.910 | 0.758 | 0.933 |
| GPT-5-mini | 0.776 | 0.786 | 0.759 | 0.846 | 0.779 | 0.834 |
| GPT-5-nano | 0.743 | 0.659 | 0.731 | 0.739 | 0.528 | 0.744 |
| GPT-4o | 0.624 | 0.644 | 0.624 | 0.650 | 0.573 | 0.653 |
| GPT-4o-mini | 0.610 | 0.646 | 0.634 | 0.529 | 0.409 | 0.489 |
| Claude-4.1-Opus | 0.806 | 0.718 | 0.794 | 0.945 | 0.851 | 0.946 |
| Claude-4-Sonnet | 0.799 | 0.734 | 0.784 | 0.891 | 0.704 | 0.904 |
| Claude-3.7-Sonnet | 0.706 | 0.656 | 0.675 | 0.888 | 0.803 | 0.870 |
| Claude-3.5-Sonnet | 0.652 | 0.615 | 0.651 | 0.836 | 0.813 | 0.688 |
| Claude-3.5-Haiku | 0.623 | 0.549 | 0.635 | 0.574 | 0.529 | 0.589 |
| Qwen2.5-VL-72B-Instruct | 0.542 | 0.586 | 0.571 | 0.575 | 0.559 | 0.609 |
| Qwen2.5-VL-7B-Instruct | 0.295 | 0.389 | 0.386 | 0.296 | 0.443 | 0.410 |
| Qwen2.5-Omni-7B | 0.501 | 0.512 | 0.448 | 0.310 | 0.307 | 0.338 |
| InternVL3-78B | 0.549 | 0.549 | 0.560 | 0.468 | 0.446 | 0.505 |
| InternVL3-8B | 0.420 | 0.431 | 0.454 | 0.372 | 0.289 | 0.364 |
| InternVL-2.5-78B | 0.552 | 0.631 | 0.635 | 0.448 | 0.404 | 0.460 |
| InternVL-2.5-8B | 0.501 | 0.466 | 0.450 | 0.219 | 0.328 | 0.300 |
| Llama-3.2-90B-Vision-Instruct | 0.495 | 0.538 | 0.535 | 0.501 | 0.363 | 0.551 |
| Llama-3.2-11B-Vision-Instruct | 0.312 | 0.446 | 0.415 | 0.304 | 0.294 | 0.340 |
| gemma-3-27b-it | 0.522 | 0.545 | 0.517 | 0.529 | 0.499 | 0.515 |
| gemma-3-12b-it | 0.454 | 0.490 | 0.476 | 0.463 | 0.446 | 0.509 |

**Music / Graph Theory**

| Model | Music Language | Music Vision | Music Vision Language | Graph Theory Language | Graph Theory Vision | Graph Theory Vision Language |
|---|---|---|---|---|---|---|
| GPT-5 | 0.806 | 0.343 | 0.764 | 0.791 | 0.684 | 0.999 |
| GPT-5-mini | 0.736 | 0.348 | 0.730 | 0.788 | 0.699 | 0.996 |
| GPT-5-nano | 0.548 | 0.328 | 0.549 | 0.768 | 0.525 | 0.990 |
| GPT-4o | 0.540 | 0.328 | 0.495 | 0.725 | 0.382 | 0.738 |
| GPT-4o-mini | 0.348 | 0.241 | 0.307 | 0.735 | 0.347 | 0.688 |
| Claude-4.1-Opus | 0.581 | 0.348 | 0.580 | 0.974 | 0.394 | 0.936 |
| Claude-4-Sonnet | 0.569 | 0.328 | 0.555 | 0.974 | 0.414 | 0.971 |
| Claude-3.7-Sonnet | 0.509 | 0.409 | 0.431 | 0.868 | 0.496 | 0.739 |
| Claude-3.5-Sonnet | 0.431 | 0.365 | 0.344 | 0.739 | 0.446 | 0.473 |
| Claude-3.5-Haiku | 0.361 | 0.301 | 0.290 | 0.561 | 0.353 | 0.471 |
| Qwen2.5-VL-72B-Instruct | 0.425 | 0.341 | 0.373 | 0.647 | 0.414 | 0.524 |
| Qwen2.5-VL-7B-Instruct | 0.282 | 0.260 | 0.280 | 0.340 | 0.307 | 0.361 |
| Qwen2.5-Omni-7B | 0.289 | 0.279 | 0.306 | 0.351 | 0.318 | 0.364 |
| InternVL3-78B | 0.411 | 0.285 | 0.361 | 0.574 | 0.430 | 0.502 |
| InternVL3-8B | 0.302 | 0.289 | 0.278 | 0.434 | 0.419 | 0.448 |
| InternVL-2.5-78B | 0.319 | 0.236 | 0.283 | 0.471 | 0.386 | 0.460 |
| InternVL-2.5-8B | 0.245 | 0.220 | 0.221 | 0.331 | 0.336 | 0.365 |
| Llama-3.2-90B-Vision-Instruct | 0.331 | 0.269 | 0.276 | 0.409 | 0.367 | 0.393 |
| Llama-3.2-11B-Vision-Instruct | 0.264 | 0.245 | 0.253 | 0.275 | 0.335 | 0.283 |
| gemma-3-27b-it | 0.430 | 0.330 | 0.354 | 0.581 | 0.339 | 0.414 |
| gemma-3-12b-it | 0.395 | 0.280 | 0.316 | 0.521 | 0.386 | 0.416 |

Figure 3: Comparison of model accuracy across different modalities in each domain.

demonstrate minimal tokenization errors. The severity of such *textual perception* limitation negatively correlates with the advantage of language inputs.

To further investigate tokenization error, we rerun the chess tasks with "gold-standard" tokenization, which separates the tokens in FEN according to human common sense. For example, if there are two pawns next to each other, the FEN will include "PP" which should intuitively be tokenized into two "P" tokens. But VLMs often suboptimally regard "PP" as a single token. We expect the models to perform better with the modified tokenizer. However, we found that Qwen2.5-VL-72B-Instruct performs almost the same (original 0.542 vs. gold-standard 0.540 accuracy). We hypothesize that two contradictory factors combined to result in such performance: although the tokenization becomes intuitively better (positive factor), the models were not trained with such tokenization (negative factor). For example, "PP" might occur a lot in the training data, and the model always uses the single "PP" token to understand FEN. When we simply replace the "PP" token with two "P" tokens, the semantics learned in the embedding of "PP" could not be directly used, while the semantics of two adjacent "P"s were not as well trained. Such observations suggest the importance of designing task-specific tokenizers and training domain-specific VLMs.

**Visual perception error.** Furthermore, our results in Fig 3 suggest parallel constraints in *visual perception*. In particular, the lack of synergistic improvement when combining vision and language input indicates that the vision modality suffers from its own perception challenges. In domains where language inputs face known tokenization difficulties, such as chemistry and chess, the failure of vision inputs to compensate for these limitations is evident across most VLMs. In domains with minimal tokenization problems, most VLMs yield better results with language than vision-language. This indicates that vision inputs are contributing negatively, which points to fundamental constraints in visual perception capabilities. In particular, the process of cutting each image into patches to feed ViTs is problematic. We found severe VLM hallucinations when vision inputs for graph theory tasks are cut near the intersection of edges. The hallucinations are particularly related to the edges and nodes involved in these intersections. A detailed example is shown in Fig 7 in the Appendix. Note that textual inputs of the graph theory tasks are presented as adjacency matrices, in which numbers were well-separated by commas, and thus there are no known tokenization issues. This could potentially explain why the vision performance is much worse than language in Fig 3.

## 5 Discussion

**Semantic equivalence.** Although we generate and curate **SEAM** with standardized tools to maximize cross-modal semantic equivalence, achieving *perfect* equivalence remains a theoretical ideal. Subtle rendering details (e.g., line thickness, fonts, layout-driven node placement) can introduce low-level perceptual differences relative to purely symbolic text, potentially interacting with model perception. Nevertheless, results in Fig 4 (Appendix B) demonstrate practical robustness to substantial visual alterations that preserve core symbolic information. Specifically, applying resolution changes, grayscale conversion, and $180°$ rotation yields only minimal performance variation (at most $\pm1.6\%$, $\pm3.1\%$, and $\pm1.9\%$, respectively), suggesting that models respond primarily to underlying content rather than superficial visual characteristics. Additionally, some domains admit minor semantic mismatches between modalities. For example, FEN includes details (e.g., castling rights, en passant availability) that are not always recoverable from a board image alone. In practice, these differences affect task performance in only a very small minority of cases.

**Limitations.** We focus exclusively on images without evaluating video or multi-image sequences, though such extensions are possible and important future work. For example, comparing chess move sequences in PGN notation with sequences of board states could assess cross-modal temporal reasoning. Our domain coverage, though carefully selected, covers only a subset of domains meeting our selection criteria; additional domains such as circuit diagrams versus SPICE netlists (Vungarala et al., 2025) could be incorporated. Our evaluation includes a limited number of models due to budget constraints for proprietary models and computational constraints for open-source models. **SEAM** will be made publicly available with a leaderboard, enabling broader community contribution and ongoing evaluation of emerging models. Future research could explore qualitative methods like report cards (Yang et al., 2024c) to address the limitations of current VLM benchmarks.

## 6 Conclusion

Our **SEAM** benchmark reveals significant modality imbalance in VLMs, which struggle to reason consistently across semantically equivalent visual and textual representations in distinct notation systems. This fundamental limitation highlights the gap between current capabilities and truly modality-agnostic AI. Moving forward, a crucial aspect of VLM development is the ability to process information regardless of its representation format. **SEAM** provides a principled framework for measuring progress toward more robust and genuinely intelligent VLMs.

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

# A    Extended Related Work

**Modality imbalance.** Despite the integration of visual and textual understanding in modern VLMs, an inherent challenge persists in achieving balanced performance across different modalities. Recent studies have uncovered significant inconsistencies between vision and language capabilities in multimodal models, showing that even state-of-the-art systems demonstrate varying levels of reliability depending on input modality (Zhang et al., 2024b; 2023c). This imbalance becomes particularly evident in complex spatial reasoning tasks, where VLMs surprisingly underperform compared to their text-only counterparts, often neglecting visual information when textual clues are available (Wang et al., 2025). The underlying mechanisms driving this phenomenon have been theoretically explored, revealing a competitive dynamic between modalities during joint training where gradient-based optimization leads to only a subset of modalities being effectively learned (Huang et al., 2022). Various approaches have emerged to address this challenge, including adaptive gradient modulation techniques that dynamically adjust the optimization pace for different modalities (Peng et al., 2022), prototype-based methods that specifically boost slower-learning modalities without interference from dominant ones (Fan et al., 2023), and explicit image-to-text conversion strategies that bridge the gap between simple and complex visual reasoning tasks (Park et al., 2025). However, existing approaches to evaluate modality imbalance lack rigorous cross-modal semantic equivalence, leaving the field without principled benchmarks to isolate and measure modality-agnostic reasoning capabilities. Further discussions on related works about domain-specific VLMs are presented in Appendix A.

**Domain-Specific VLMs.** While general-purpose VLMs have shown remarkable capabilities across diverse tasks, a growing trend has emerged in developing domain-specialized vision-language models that focus on particular fields requiring expert knowledge. NOTA (Tang et al., 2025) represents one such effort in the music domain, bridging the gap between two-dimensional score images and one-dimensional symbolic notation through a dedicated multimodal music notation dataset. In the medical domain, MedVInT (Zhang et al., 2023d) and Med-PaLM (Tu et al., 2024) leverage specialized medical visual-textual pretraining to enhance clinical reasoning and diagnosis. For scientific literature, SciMMIR (Wu et al., 2024) focuses on scientific diagram understanding and multimodal information retrieval. Other domain-specific efforts include ChartLlama (Han et al., 2023) for chart interpretation, Mplug-DocOwl (Ye et al., 2024) for document understanding, and LayoutGPT (Feng et al., 2023) for graphic design. These specialized models often demonstrate superior performance in their target domains compared to general-purpose VLMs, highlighting the importance of domain-specific knowledge and training data in addressing specialized visual reasoning tasks.

# B    Additional Results

| Modality | Extraction Model | Average Accuracy | Agreement |
|---|---|---|---|
| Language | Qwen2.5-7B-Instruct | 0.635 | |
| | Qwen2.5-72B-Instruct | 0.635 | 0.997 |
| Vision | Qwen2.5-7B-Instruct | 0.482 | |
| | Qwen2.5-72B-Instruct | 0.481 | 0.998 |
| Vision-Language | Qwen2.5-7B-Instruct | 0.627 | |
| | Qwen2.5-72B-Instruct | 0.627 | 0.998 |

Table 2: GPT-4o results with different final answer extractors.

## B.1    Semantic Equivalence

Admittedly, achieving perfect semantic equivalence across modalities presents theoretical challenges, our benchmark design implements rigorous controls to ensure comparable

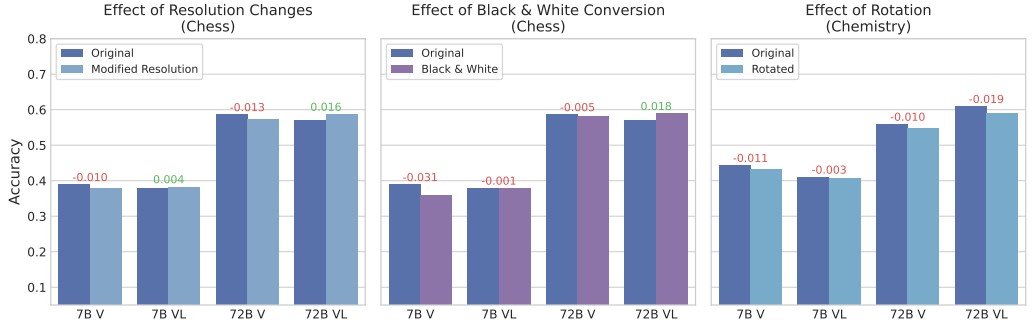

Figure 4: Effect of performance from visual transformations on Qwen2.5-VL models.

information content between visual and textual representations. To validate the robustness of our approach, we conduct experiments examining how visual transformations affect model performance, as shown in Figure 4. We systematically tested three common image transformations: resolution changes in chess diagrams, black-and-white conversion of chess boards, and 180-degree rotation of molecular structures on Qwen 2.5 models.

Our results demonstrate minimal performance variations across these modifications. For resolution changes, we observe differences of no more than ±1.6% in accuracy. Similarly, black-and-white conversion produces maximum deviations of ±3.1%, while molecular rotation yields changes within ±1.9%. The stability of performance across these transformations supports our claim that models are responding to the underlying semantic content rather than superficial visual properties. This is particularly notable given that these modifications substantially alter the pixel-level representation while preserving the symbolic information content.

## B.2 Cross-model Agreement

The cross-model agreement analysis on semantically equivalent tasks reveals striking modality-dependent variations in model consistency. When processing the same problems in language (left panel), models demonstrate high within-family agreement (GPT variants: 0.79-0.86, Claude variants: 0.77-0.82) and moderate cross-family agreement (0.64-0.75), suggesting that textual-symbolic reasoning elicits relatively consistent responses across models. However, this consistency deteriorates markedly in the vision modality (center panel), where agreement scores drop substantially, indicating that visual-spatial processing of the same semantic content produces more divergent model behaviors. This aligns with our finding that vision frequently underperforms language despite semantically equivalent information being provided.

## B.3 Error Analysis

Our analysis revealed two distinct failure patterns in text versus vision modalities. These examples demonstrate fundamental limitations in how each modality handles structured information.

**Carbon counting:** The text-only system incorrectly tokenized the SMILES notation as:

**SMILES representation.**

```
Nc1nccc(Oc2cc(F)ccc2F)c1I
```

**Tokenized sequence.**

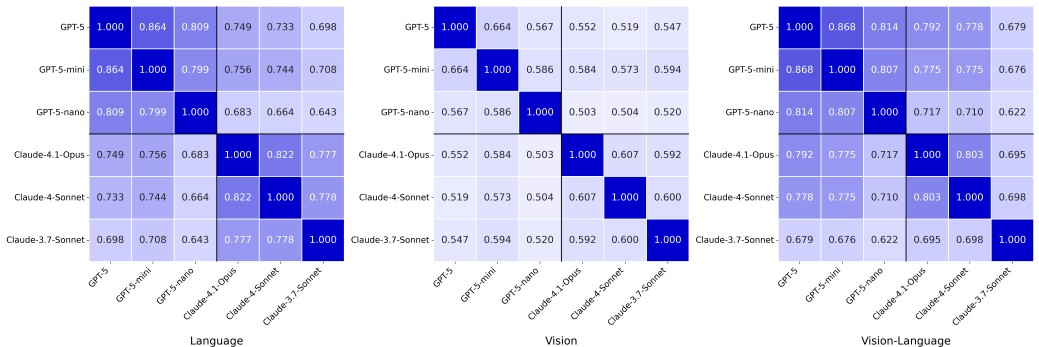

Figure 5: Cross-model agreement on semantically equivalent tasks presented in different modalities. Correlation matrices show pairwise agreement scores between GPT-5, GPT-5-mini, GPT-5-nano, Claude-4.1-Opus, Claude-4-Sonnet, and Claude-3.7-Sonnet models when solving the same problems in language (left), vision (center), and vision-language (right) formats, with darker blue indicating higher agreement.

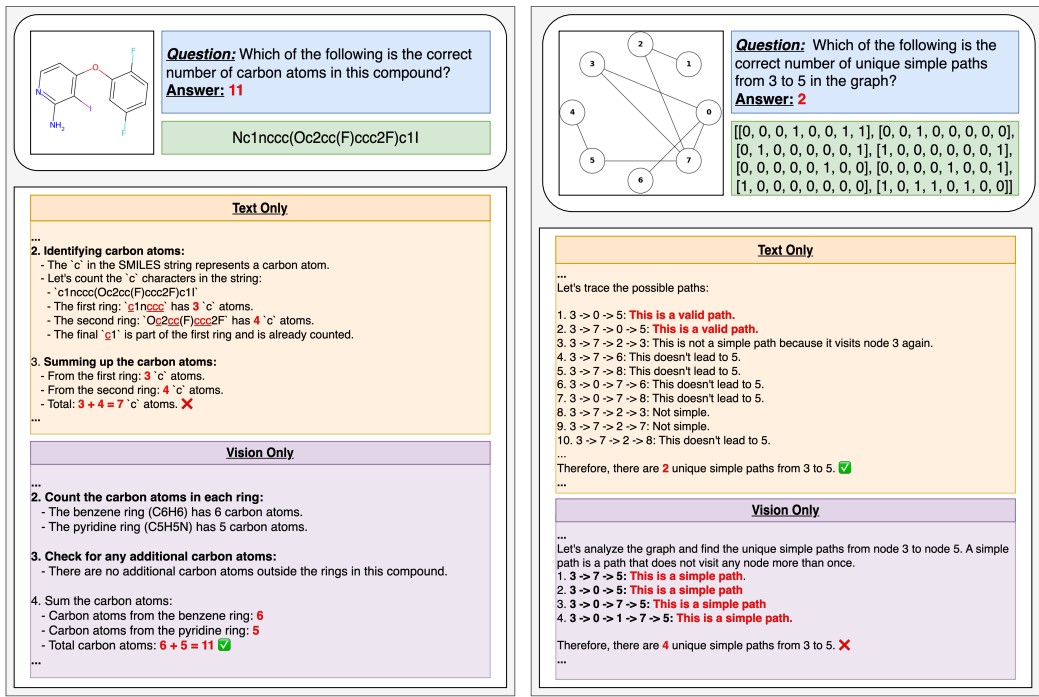

Figure 6: Case-study of text-based versus vision-based processing in VLMs, showing two error cases: (left) Qwen2.5-VL-72B-Instruct incorrect counting of carbon atoms in a chemical structure due to SMILES tokenization, and (right) gemma-3-12b-it hallucinated graph paths in a simple path counting problem. Both examples show discrepancies between text-only and vision-only representatinos.

```
[`N', `c', `1', `n', `ccc', `(O', `c', `2', `cc', `(F', `)', `ccc', `2', `F', `)c', `1',
 `I']
```

This results in counting only 7 carbon atoms instead of the correct 11. This error stems from text systems struggling with context-dependent character interpretation in specialized notations where 'c' signifies aromatic carbon. In contrast, the vision-only approach correctly identified 11 carbon atoms by properly recognizing the benzene ring (6 carbons) and pyridine ring (5 carbons) in the 2D molecular structure.

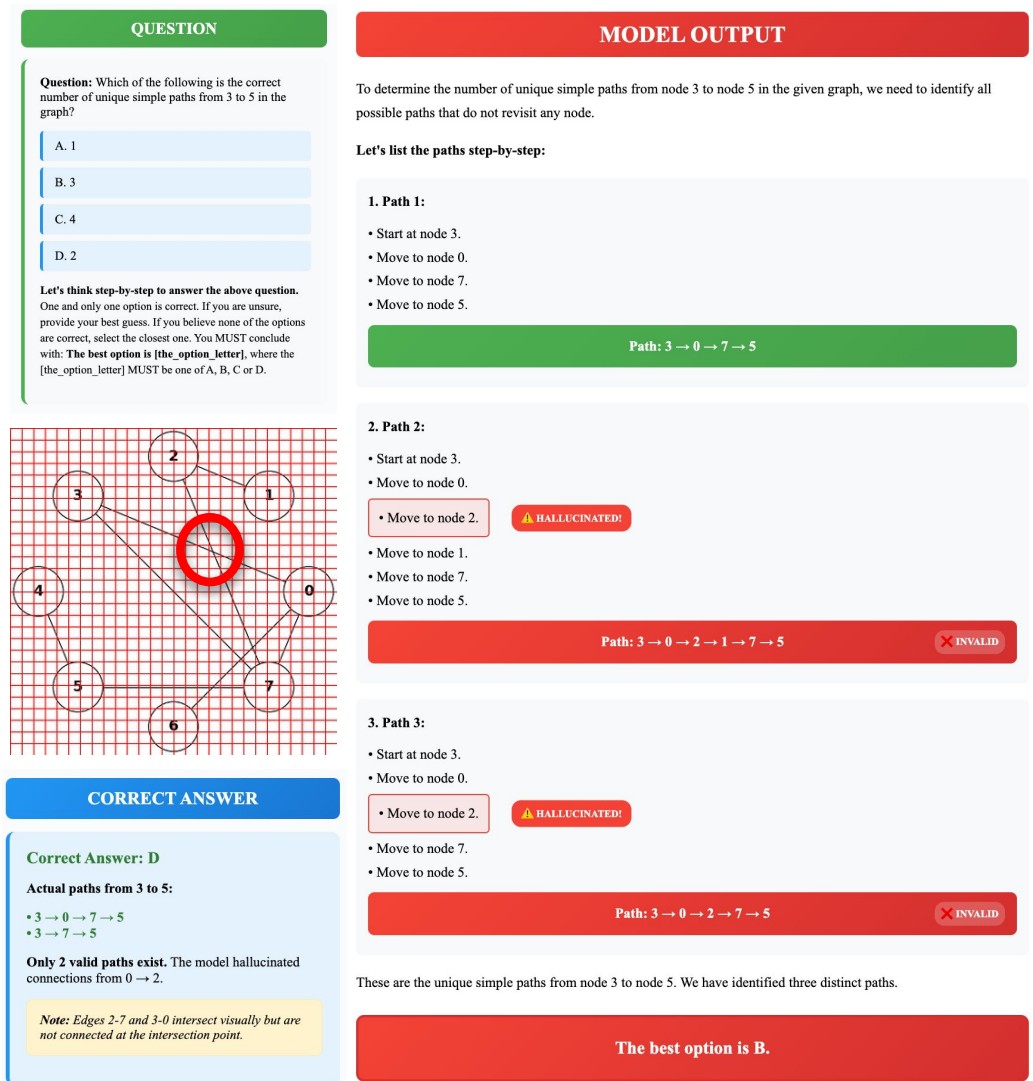

Figure 7: Visual perception error example.

**Unique simple path counting:** When analyzing a simple undirected graph, the text-only approach correctly identified exactly 2 unique simple paths from node 3 to node 5 through path enumeration. However, the vision-only approach erroneously counted 4 paths by hallucinating edges that don't exist in the graph ($0 \rightarrow 5, 0 \rightarrow 1, 1 \rightarrow 7$). This suggests vision systems may struggle with precise spatial reasoning in graphs, potentially interpolating connections based on spatial proximity rather than actual edge connections.

These examples highlight complementary strengths and weaknesses across input modalities. Text-based systems excel at following explicit rules but may fail with specialized notation requiring contextual understanding. Vision-based systems can interpret visual structures holistically but may hallucinate relationships based on visual proximity. Our findings suggest potential benefits from multimodal approaches that cross-validate across modalities to mitigate modality-specific errors.

## B.4 Final Answer Extractor

While VLMs often generate verbose rationales when prompted for multi-choice answers, we need to extract their definitive selection of option A, B, C, or D robustly. We evaluate two

Table 3: Hyperparameter settings for benchmark creation and model inference.

| Category | Parameter | Value |
|---|---|---|
| **Graph Tasks** | Node Limits | Min: 6, Max: 9 |
| | Edge Limits | Min: 5, Max: 20 |
| | Path Count Offset | $\pm \lceil 0.1 \times \text{edge count} + 1 \rceil$ from correct count |
| | Image Size | $400 \times 400$ px |
| | Samples | 200 per task |
| **Chemistry Tasks** | Carbon Offset | $\pm 3$ from correct count |
| | Hydrogen Count Offset | $\pm 3$ from correct count |
| | Weight Count Offset | $\pm 20\%$ of correct molecular weight |
| | Image Size | $400 \times 400$ px |
| | Samples | 200 per task |
| **Chess Tasks** | Max Puzzle Rating | 1200 |
| | CP Offset | $\pm 300$ centipawns from correct evaluation |
| | Image Size | $400 \times 400$ px |
| | Samples | 200 per task |
| **Music Tasks** | Measure Limits | Min: 24, Max: 48 measures |
| | Note Count Offset | $\pm 3$ from correct count |
| | Measure Count Offset | $\pm 3$ from correct count |
| | Image Size | $600 \times 600$ px |
| | Samples | 200 per task |
| **Model Inference** | Temperature Inference | Model Default |
| | Temperature Extraction | 0 |
| | Max New Tokens | 8192 |
| | Max Model Length | 16384 |

potential extractors: Qwen2.5-7B-Instruct and Qwen2.5-72B-Instruct, with vast difference in the scale of parameters, comparing their extraction reliability on GPT-4o outputs across all modalities. As shown in Table 2, both extractors achieved near-identical results with exceptionally high agreement across all modalities. The consistency between the 7B and 72B models indicates that answer extraction is a relatively straightforward task that hardly benefit from additional model capacity. Therefore, we adopt the more computationally efficient Qwen2.5-7B-Instruct as our standard extractor for all experiments, assigning a special token Z in cases where no valid option could be extracted, rather than making random assignments that might artificially inflate performance metrics.

## C Reproducibility

### C.1 Hyperparameter Settings

Our benchmarking framework involves carefully designed hyperparameters across different task categories and model inference settings. Table 3 presents the comprehensive hyperparameter configuration used in our experiments for reproducibility purposes.

### C.2 Prompt Templates

**Language-only Prompt Template.**

```
{notation_name}: {notation}

{instructions}

A. {option_a}
B. {option_b}
C. {option_c}
D. {option_d}

Let's think step-by-step to answer the above question.
One and only one option is correct. If you are unsure, provide your best guess.
If you believe none of the options are correct, select the closest one.
You MUST conclude with: The best option is [the_option_letter],
where the [the_option_letter] MUST be one of A, B, C or D.
```

**Vision-only Prompt Template.**

```
{image}

{instructions}

A. {option_a}
B. {option_b}
C. {option_c}
D. {option_d}

Let's think step-by-step to answer the above question.
One and only one option is correct. If you are unsure, provide your best guess.
If you believe none of the options are correct, select the closest one.
You MUST conclude with: The best option is [the_option_letter],
where the [the_option_letter] MUST be one of A, B, C or D.
```

**Language & Vision Prompt Template.**

```
{image}

{notation_name}: {notation}

{instructions}

A. {option_a}
B. {option_b}
C. {option_c}
D. {option_d}

Let's think step-by-step to answer the above question.
One and only one option is correct. If you are unsure, provide your best guess.
```

```
If you believe none of the options are correct, select the closest one.
You MUST conclude with: The best option is [the_option_letter],
where the [the_option_letter] MUST be one of A, B, C or D.
```

**Prompt Template for Final Answer Extraction.**

```
Here is the complete predicted answer for a multiple-choice question\

***{prediction}***

Your task: Extract the final answer (the best option) from the text above.
Ignore the reasoning process and any inconsistence in the above complete predicted
answer.
It is usually in the format 'The best option is [letter]' at the end of the complete
predicted answer.
If found, reply with the letter 'A', 'B', 'C', or 'D'.
Otherwise, reply with 'Z'.
```

## C.3    Task Design Details

Our task designs focus on creating specialized problems that test the ability of models to understand information presented across different modalities. Each domain requires careful construction of semantically equivalent representations while ensuring appropriate difficulty levels and clear evaluation metrics.

### C.3.1    Chess

Chess offers an ideal domain for testing modality equivalence due to its well-established symbolic notation systems and visual representations via standard tools. We developed four chess-based tasks that assess different aspects of chess understanding and reasoning, ensuring that questions can be answered through either visual board inspection or formal chess notation analysis.

**Fork Detection.** The fork detection task evaluates a model's ability to identify pieces creating tactical *forks* on the chess board. A fork occurs when a single piece simultaneously attacks two or more opponent pieces. We constructed this task using the python-chess library to analyze positions from the Lichess puzzle database. For each position, we load the FEN (Forsyth–Edwards Notation) string and analyze all pieces on the board to identify those attacking multiple opponent pieces simultaneously using *board.attacks()* to count the number of opponent pieces under attack from each piece from python-chess library. A random piece creating a fork is selected as the correct answer, while three random non-forking pieces are sampled as incorrect options. All incorrect options reference legitimate pieces of the appropriate color on the board, making the task require genuine tactical understanding rather than simple pattern matching.

**Legal Move Discrimination.** The legal move discrimination task evaluates whether models can distinguish legal from illegal moves in complex positions. Chess positions frequently contain moves that appear valid but are illegal due to pins, checks, or other tactical constraints. We constructed this task by loading chess positions from FENs and identifying legal moves using the *board.legal_moves* generator from the python-chess library. We then generate plausible-looking illegal moves from appropriate pieces but to invalid destinations. These illegal moves fall into several categories: moves that would leave the king in check, moves of the correct piece color but to illegal destinations, and moves that appear geometrically plausible but violate piece movement rules. For each position, we select one random legal move and three illegal moves, presenting the position and candidate moves across all modalities.

**Puzzle Solving.** The puzzle solving task challenges models to find the best move in tactical positions drawn from the Lichess puzzle database. We carefully selected puzzles with a maximum ELO rating to ensure appropriate difficulty for current models. For each puzzle, we parse the FEN position and the first move from the puzzle sequence, then apply this move to reach the position where the model must find the best continuation. The best move, provided by the dataset, is presented alongside three randomly selected legal but suboptimal moves as multiple-choice options obtained from python-chess library.

**Position Evaluation.** The position evaluation task assesses a model's ability to judge the relative advantage in a chess position. Professional chess engines like Stockfish quantify advantage using centipawn values, where positive values indicate an advantage for White and negative values for Black. We selected positions from a pre-evaluated Lichess database where positions had moderate advantages to avoid trivial cases. For each position, we created four evaluation options by taking the correct centipawn value and applying offsets to the correct answer. This task is particularly challenging as it requires holistic assessment of multiple factors including material balance, piece activity, king safety, and pawn structure that transcend simple piece counting to correctly evaluate positions.

All chess tasks utilize python-chess library's SVG rendering capabilities through *chess.svg.board()* to generate visualizations, which are then converted to PNG images using

Cairosvg. Positions are presented with appropriate board orientation, with board flipped when Black to move and with coordinate notations to ensure complete information is available in the visual modality. For each task, we carefully selected 200 samples to create a balanced and challenging dataset.

### C.3.2 Chemistry

Molecular chemistry provides another appropriate test domain for modality equivalence due to its dual representation systems: visual structural diagrams and symbolic SMILES notation. We developed four chemistry-based tasks that evaluate understanding of molecular properties across different modalities, ensuring each task can be solved through either visual inspection of molecular diagrams or analysis of SMILES notation.

**Carbon Counting.** The carbon counting task tests a model's ability to accurately count the number of carbon atoms in organic molecules. We selected molecules from the ChemQA dataset with substantial carbon content to create appropriately challenging problems. Using RDKit, we parse the SMILES notation of each molecule and programmatically identify the exact carbon atom count to yield the correct answer, with incorrect options differing by multiples of offset.

**Hydrogen Counting.** The hydrogen counting task is similar to the carbon counting one. Using RDKit's *GetTotalNumHs()* method, we calculate the total hydrogen count for each molecule, including both explicit and implicit hydrogens. This task requires understanding of organic chemistry valence rules to correctly infer hydrogen positions that may not be explicitly shown in the molecular diagram or directly encoded in the SMILES string.

**Molecular Weight Calculation.** The molecular weight calculation task assesses a model's ability to estimate the molecular mass of chemical compounds. We use RDKit's *Descriptors.MolWt()* function to calculate the exact molecular weight of each compound based on its atomic composition. For each molecule, we generate four options where the correct answer is the actual molecular weight and incorrect options differ by a percentage-based offset.

**Molecular Caption Matching.** The molecular caption matching task evaluates semantic understanding of molecules by requiring models to identify the most accurate description for a given molecule. We curated a set of expert-written molecule descriptions from the ChemQA dataset, where each description explains a molecule's structure, function, or biological activity. For each target molecule, we present its structure along with four possible descriptions, only one of which correctly describes the molecule. To ensure challenging distractors, we use the Multilingual-e5-large-instruct embedding model to identify semantically similar but incorrect captions based on cosine similarity.

The visual representations of all chemistry tasks are generated using RDKit's *Draw.MolToImage()* function with consistent size parameters. For each task, we carefully selected 200 samples to create a balanced and challenging dataset.

### C.3.3 Music

Musical notation provides a complex test domain for modality equivalence, as it combines visual symbols with structured temporal patterns. We developed four music-based tasks that evaluate understanding of musical compositions across different modalities, ensuring each task can be solved through either visual analysis of sheet music or interpretation of the symbolic ABC notation format.

**Note Counting.** The note counting task assesses a model's ability to accurately count specific note occurrences in musical compositions. Using the music21 library, we parse ABC notation from the Irishman dataset and systematically count occurrences of each note type from A to G using the *score.flat.getElementsByClass('Note')* method. For each piece, we randomly select a target note that appears in the composition and create multiple-choice

options where the correct answer represents the actual note count, with incorrect options differing by an offset that scales based on the correct count.

**Measure Counting.** The measure counting task tests a model's ability to identify the total number of measures in a musical composition. We use music21's parsing to count measures from the ABC notation, applying the *part.getElementsByClass('Measure')* method to identify distinct measures in the score. As with the note counting task, the offset for incorrect options scales with the distance from the correct answer, making the task more difficult for pieces with ambiguous measure counts or complex repeat structures that might confuse measure counting.

**Musical Form Identification.** The musical form identification task evaluates a model's understanding of compositional structure. Using a curated dataset of musical pieces with labeled forms from the music theory dataset[1], we present models with sheet music or ABC notation and ask them to identify the correct musical form from options including Only One Section, Through Composed, Compound Binary, Compound Ternary, and American Popular forms. This task requires understanding of how musical sections relate to each other and recognition of structural patterns that define different compositional forms, testing higher-level music theory knowledge.

**Rhythm Pattern Detection.** The rhythm pattern detection task examines a model's ability to identify specific rhythmic patterns within a composition. Using custom pattern matching regular expressions that analyze ABC notation, we identify measures containing specific dotted note patterns (including dotted sixteenth, dotted eighth, dotted quarter, or dotted half notes). For each composition, we select a measure containing the target rhythm pattern and create options with three measures that don't contain the pattern to make up the false answers.

All music tasks are presented in both standard five-line staff format generated using music21's rendering capabilities and symbolic ABC notation format. The sheet music images are standardized to 600×600 pixel square images, ensuring consistent visual representation. We carefully filtered and balanced each task category to create a comprehensive benchmark of 200 samples for evaluating music understanding across modalities.

### C.3.4  Graph

We present a series of stochastic algorithms for generating graph-related tasks. Each algorithm dynamically creates suitable graphs with specific constraints to ensure appropriate difficulty levels and clear conceptual focus.

The framework follows a general pattern: (1) Generate candidate graphs with controlled parameters, (2) Filter graphs to ensure they satisfy task-specific constraints, (3) Generate correct solutions, plausible distractors, and incorrect options, and (4) Format and return the complete task with answer options.

Each specific task generator (Cycle Detection, Path Counting, Path Existence, BFS Traversal) implements this framework with task-specific constraints and choice generation logic.

---

[1]https://huggingface.co/datasets/Seeker38/music_abc_notation_with_music_theory

---

**Algorithm 1:** Generate Cycle Detection Task

---

Initialize max_attempts $\leftarrow$ 50
**for** *attempt* $= 1$ **to** *max_attempts* **do**
    Generate random directed graph $D$ with few nodes
    Convert $D$ to undirected graph $G$
    **if** *D not connected* **then**
        **continue**
    **end**
    **if** *D has no cycles* **then**
        **continue**
    **end**
    directed_cycles $\leftarrow$ all simple cycles in $D$
    undirected_cycles $\leftarrow$ all simple cycles in $G$
    **if** *no new cycles in undirected graph* **then**
        **continue**
    **end**
    ▷ Design answer choices
    Correct $\leftarrow$ random cycle from directed_cycles
    Confusion $\leftarrow$ cycle in $G$ but not in $D$
    ▷ Generate incorrect answers
    Incorrect $\leftarrow$ two paths that:
        - Are not part of any cycle
        - Cannot form a cycle
        - Are distinct
    **if** *failed to generate any answer* **then**
        **continue**
    **end**
    **return** $D$, $G$, Correct, Confusion, Incorrect
**end**
**return** failure

---

---

**Algorithm 2:** Generate Path Counting Task

---

**begin**
$\quad$ Initialize max_attempts $\leftarrow$ 50
$\quad$ **for** *attempt* = 1 **to** *max_attempts* **do**
$\qquad$ Generate random undirected graph *G* with few nodes
$\qquad$ **if** *G not connected* **then**
$\qquad\qquad$ | **continue**
$\qquad$ **end**
$\qquad$ **if** *too few nodes in G* **then**
$\qquad\qquad$ | **continue**
$\qquad$ **end**
$\qquad$ Create and shuffle all possible source-target pairs
$\qquad$ **foreach** (*source, target*) *in shuffled pairs* **do**
$\qquad\qquad$ **if** *no path exists from source to target* **then**
$\qquad\qquad\qquad$ | **continue**
$\qquad\qquad$ **end**
$\qquad\qquad$ Count all simple paths from *source* to *target*
$\qquad\qquad$ **if** $1 < path\_count < 10$ **then**
$\qquad\qquad\qquad$ $\triangleright$ Generate answer options
$\qquad\qquad\qquad$ *correct_answer* $\leftarrow$ *path_count*
$\qquad\qquad\qquad$ *offset* $\leftarrow \lceil 0.1 \times edge\_count + 1 \rceil$
$\qquad\qquad\qquad$ *options* $\leftarrow$ empty list
$\qquad\qquad\qquad$ **for** *delta* $= -offset$ **to** *offset* **do**
$\qquad\qquad\qquad\qquad$ **if** *delta* $\neq 0$ *and path_count + delta* $> 0$ **then**
$\qquad\qquad\qquad\qquad\qquad$ | Add *path_count + delta* to *options*
$\qquad\qquad\qquad\qquad$ **end**
$\qquad\qquad\qquad$ **end**
$\qquad\qquad\qquad$ **while** $|options| < 3$ **do**
$\qquad\qquad\qquad\qquad$ *offset* $\leftarrow$ *offset* $+ 1$
$\qquad\qquad\qquad\qquad$ Add viable options using new offset
$\qquad\qquad\qquad$ **end**
$\qquad\qquad\qquad$ **if** $|options| > 3$ **then**
$\qquad\qquad\qquad\qquad$ Sort *options* by distance from *path_count*
$\qquad\qquad\qquad\qquad$ Keep only closest 3 options
$\qquad\qquad\qquad$ **end**
$\qquad\qquad\qquad$ **return** *G, source, target, path_count, options*
$\qquad\qquad$ **end**
$\qquad$ **end**
$\quad$ **end**
$\quad$ **return** failure
**end**

---

---

**Algorithm 3:** Generate Path Existence Task

---

**begin**
    Initialize max_attempts ← 50
    **for** *attempt* = 1 **to** *max_attempts* **do**
        Generate random directed graph *D* with few nodes
        Convert *D* to undirected graph *G*
        **if** *D not connected or too few nodes* **then**
            **continue**
        **end**
        Create and shuffle all possible source-target pairs
        **foreach** (*source*, *target*) *in shuffled pairs* **do**
            **if** *no path exists from source to target in D* **then**
                **continue**
            **end**
            Get directed paths in *D* and undirected paths in *G*
            **if** *too few or too many paths* **then**
                **continue**
            **end**
            **if** *any undirected path differs from all directed paths* **then**
                *source_node* ← *source*
                *target_node* ← *target*
                ▷ Generate answers
                *correct* ← random directed path from *source* to *target*
                *tricky* ← undirected path not in directed paths
                ▷ Generate incorrect answers
                *incorrect* ← empty list
                **for** *up to 50 attempts* **do**
                    **if** |*incorrect*| = 2 **then**
                        **break**
                    **end**
                    ▷ Try different strategies to generate invalid paths
                    *non_path* ← one of:
                        1. Path with invalid edges
                        2. Path with repeated nodes
                        3. Random sequence of nodes
                    **if** *non_path is valid and not duplicate* **then**
                        Add *non_path* to *incorrect*
                    **end**
                **end**
                **if** *correct* ≠ *None and tricky* ≠ *None and* |*incorrect*| = 2 **then**
                    **return** *D, G, correct, tricky, incorrect*
                **end**
            **end**
        **end**
    **end**
    **return** failure
**end**

---

---

**Algorithm 4:** Generate BFS Traversal Task

---

**begin**
  ▷ Generate suitable graph and BFS levels
  **for** *attempt* = 1 **to** *max_attempts* **do**
    Generate random undirected graph *G*
    **if** *G not connected or has too few nodes* **then**
      **continue**
    **end**
    **foreach** *start_node in shuffled nodes* **do**
      *level_groups* ← BFS levels from *start_node*
      **if** *fewer than 3 levels or unsuitable level sizes* **then**
        **continue**
      **end**
      *correct_order* ← flattened *level_groups* (nodes sorted within levels)
      **return** *G*, *start_node*, *level_groups*, *correct_order*
    **end**
  **end**
  ▷ Generate answer options
  *confusing_distractor* ← None
  **for** *several attempts* **do**
    Copy *level_groups* to *distractor_levels*
    Swap nodes between non-root levels
    *distractor_order* ← flattened *distractor_levels*
    **if** *distractor_order* ≠ *correct_order* **then**
      *confusing_distractor* ← *distractor_order*
      **break**
    **end**
  **end**
  *incorrect_options* ← empty list
  **for** *several attempts* **do**
    **if** |*incorrect_options*| = 2 **then**
      **break**
    **end**
    ▷ Try shuffling level structure
    Copy and shuffle non-root levels while keeping root fixed
    *incorrect_order* ← flattened shuffled levels
    **if** *incorrect_order is valid and unique* **then**
      Add to *incorrect_options*
    **end**
  **end**
  ▷ Format and return results
  Format all traversal options with level grouping
  Combine into *option_list* and shuffle
  *correct_idx* ← index of correct option
  **return** *G*, *option_list*, *correct_idx*, *level_sizes*
**end**

---

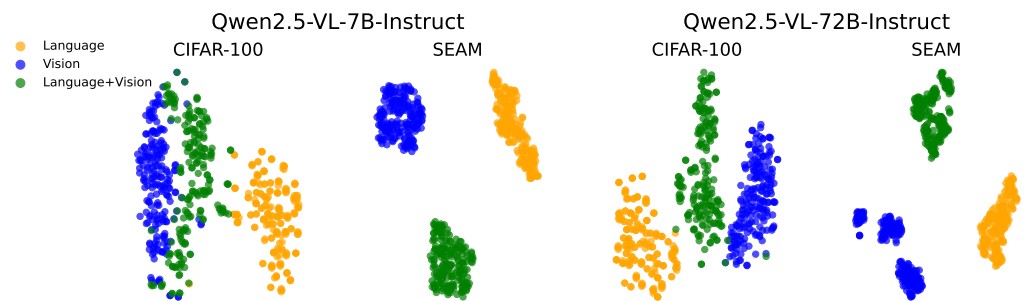

Figure 8: T-SNE visualization of modality-specific embeddings from Qwen models.

## D   Internal Representation Alignment

**SEAM** also enables us to investigate how VLMs process semantically equivalent information across different input formats internally. A fundamental capability of robust VLMs is to create unified representations across modalities that align on semantic content rather than cluster by input format (Pandey et al., 2022; Zhang et al., 2023a). Following insights from Huang et al. (2024) that effective VLMs should narrow the modality gap in deeper layers, we examine the final layer representations of both Qwen2.5-VL-7B-Instruct and Qwen2.5-VL-72B-Instruct models to investigate whether current VLMs achieve this capability on our challenging benchmark.

We include CIFAR-100 (Krizhevsky et al., 2009) as a comparison baseline because it represents a simple and well-established dataset, serving as an approximate upper bound for cross-modal alignment. We extract embeddings from three input modalities: language-only, vision-only, and multimodal (language plus vision) for both datasets. These embeddings are then projected into two-dimensional space using T-SNE (Van der Maaten & Hinton, 2008) for visualization and analysis. Fig 8 presents these T-SNE visualizations from both model scales. For CIFAR-100, we observe that the models demonstrate integrations of embeddings across modalities, indicating fairly successful cross-modal alignment for simple categorical information. In contrast, our **SEAM** benchmark reveals persistent modality gaps at the final layer in both models, with embeddings from all three modalities forming distinct clusters with minimal overlap. This disparity in separation magnitude suggests the issue extends beyond any natural tendency to preserve modality-specific information for output purposes, as both datasets would show similar patterns if that were the only factor. Instead, this separation demonstrates that despite impressive performance on standard benchmarks, current VLMs struggle to form unified representations when reasoning over semantically equivalent information presented in fundamentally different notation systems.

### D.1   Embedding Extraction

For the Qwen2.5-VL-7B-Instruct and Qwen2.5-VL-72B-Instruct models analyzed, we extract embeddings from each transformer layer $l \in \{1, 2, ..., L\}$ in the language decoder part, where $L$ represents the total number of layers. Given an input $i$ in modality $m \in \{\text{language}(L), \text{vision}(V), \text{vision-language}(VL)\}$, we extract the hidden states $h_{i,m}^l \in \mathbb{R}^{s_m \times d}$, where $s_m$ is the sequence length for modality $m$ and $d$ is the hidden dimension.

To obtain a single representative embedding for each input-modality pair, we apply mean pooling across all tokens:

$$\bar{h}_{i,m}^l = \frac{1}{s_m} \sum_{j=1}^{s_m} h_{i,m,j}^l \tag{1}$$

This approach captures comprehensive representations for both image and text inputs, rather than relying solely on special tokens like [CLS] or [IMG], which might not fully encapsulate modality-specific information in intermediate layers (Jiao et al., 2024).

### D.2 Cross-Modal Representation Analysis

#### D.2.1 T-SNE Visualization

We run T-SNE on the mean-pooled embeddings to visualize the embedding spaces across modalities. For the models involved, we extract embeddings from the final layer for inputs from CIFAR-100 and our **SEAM** benchmark in three modalities: language, vision, and language+vision. For the CIFAR-100 dataset, we use class labels as the language input. For our **SEAM** benchmark, we use the pure symbolic representation without further guidance prompts as the language input. The T-SNE algorithm is then applied with a perplexity of 30 and 1,000 iterations to obtain a two-dimensional projection that preserves local neighborhood relationships.

