# OpenReview forum: "SEAM: Semantically Equivalent Across Modalities Benchmark for Vision-Language Models"
_colmweb.org/COLM/2025/Conference — COLM 2025_

### Official Review · Reviewer_NLx2 · 2025-05-01

**Rating:** 6
**Confidence:** 4
**Ethics Flag:** 1

**Summary:**

This paper investigates the performance inconsistency of vision-language models (VLMs) when identical semantic content is presented in different input modalities, such as visual versus textual. The study reveals that current VLMs suffer a significant performance degradation when a piece of information is provided in visual modality as opposed to text, even when the semantic content remains the same. This raises notable questions about multi-modal understanding capabilities of these models—particularly their robustness and consistency across modalities.

**Reasons To Accept:**

- The paper tackles a timely and important question: how reliable are VLMs when the modality changes but semantic content remains unchanged?
- It highlights a notable performance gap when switching to the vision modality, which raises critical concerns around robustness and modality invariance in VLMs.

**Reasons To Reject:**

- The paper lacks a quantitative error analysis to determine why vision modality leads to weaker performance. Is it a result of perception issues or reasoning limitations? Potential TODOs:
    - Test whether vision inputs could be reliably transformed into equivalent textual representations using existing VLM capabilities—an analysis that could significantly complement the core findings.
    - Test two-stage VQA solving architectures (e.g., VLM description followed by LLM reasoning [1]), which is increasingly common in practice and may show a different performance profile.
- There is no comparison of  VLMs & their corresponding LLMs on the textual version of SEAM benchmark, leaving the cause of modality-induced degradation unclear.

[1] Prism: A Framework for Decoupling and Assessing the Capabilities of VLMs

---

> ### Author Response · Authors · 2025-06-03
>
> Thank you for the recognition of our work. We appreciate your suggestions for deeper quantitative error analysis, two-stage VQA architectures, and VLM-LLM comparisons. While these are excellent directions that could provide valuable insights into the underlying causes of modality imbalances, we respectfully note that such comprehensive causal analysis and improvement strategies extend beyond the scope of our current benchmark contribution.
>
> Our primary goal with SEAM is to establish a rigorous evaluation framework that can reliably measure cross-modal consistency using semantically equivalent representations, which was previously difficult to assess with existing benchmarks. The detailed investigation of *why* these gaps occur (perception vs. reasoning limitations) and *how* to address them (through two-stage architectures, specialized training, etc.) represents substantial independent research directions as future works that would benefit from the standardized testbed we provide.

---

> > ### Comment · Reviewer_NLx2 · 2025-06-05
> >
> > Thank you for the response, I will keep my initial score.

---

### Official Review · Reviewer_yu5A · 2025-05-09

**Rating:** 6
**Confidence:** 4
**Ethics Flag:** 1

**Summary:**

This paper introduces SEAM, a novel benchmark for evaluating vision-language models (VLMs) on their ability to reason consistently across semantically equivalent representations in different modalities. Unlike existing benchmarks that use OCR-derived image-text pairs, SEAM leverages domains with standardized notation systems in both visual and textual formats: chess (FEN vs. board images), chemistry (SMILES vs. structural diagrams), music (ABC notation vs. sheet music), and graph theory (adjacency matrices vs. node-edge diagrams).
The authors evaluate 13 state-of-the-art VLMs on this benchmark and discover significant modality imbalances across all models. They find a strong correlation between cross-modal answer agreement and overall accuracy, suggesting that models with better modality alignment tend to perform better generally.

**Questions To Authors:**

Have you considered extending SEAM to include other domains with standardized notation systems, such as circuit diagrams vs. SPICE netlists (as mentioned in your limitations)?

For the robustness tests of semantic equivalence (Fig. 5), have you considered more comprehensive transformations beyond resolution changes, B&W conversion, and rotation?

The paper mentions that cross-modal answer agreement correlates with accuracy. Have you investigated whether explicitly optimizing for cross-modal agreement during training could improve VLM performance?

You mention that tokenization errors are a significant issue, particularly for SMILES strings. Have you considered experimenting with specialized tokenizers for domain-specific notations?

The internal representation analysis shows persistent modality gaps in embedding space. Have you investigated at which layer in the model architecture this separation begins to emerge?

**Reasons To Accept:**

Novel evaluation paradigm: SEAM provides a fundamentally new way to evaluate VLMs by testing their reasoning across distinct notation systems rather than just OCR-derived text, addressing a significant gap in existing benchmarks.

Rigorous benchmark design: The authors carefully select domains with standardized notation systems, ensuring semantic equivalence while providing challenging reasoning tasks that require domain-specific understanding.

Important empirical findings: The discovery that cross-modal answer agreement strongly correlates with overall accuracy provides actionable insights for future VLM development.

Thorough model evaluation: The paper evaluates a comprehensive set of 13 VLMs spanning both proprietary and open-source models of various sizes, providing valuable comparative analysis.

Insightful error analysis: The detailed investigation of modality-specific failures (e.g., tokenization errors, visual perception limitations) helps identify specific areas for improvement in current VLMs.

Internal representation analysis: The paper goes beyond surface-level performance metrics to examine how VLMs internally process semantically equivalent information across modalities.

**Reasons To Reject:**

Limited domain coverage: While the four selected domains are well-justified, they represent only a subset of possible domains with dual representation systems, potentially limiting generalizability.

Task diversity limitations: Each domain has only 4 tasks with 200 samples each, which may not capture the full range of reasoning capabilities required in those domains.

Complexity control concerns: The paper doesn't fully address how well the benchmark controls for inherent complexity differences between modalities (a textual representation might inherently be more/less complex than its visual counterpart).

Limited investigation of improvement strategies: While the paper identifies limitations in current VLMs, it offers relatively few concrete suggestions for how to address modality imbalance issues.

---

> ### Author Response · Authors · 2025-06-03
>
> Thank you for your thoughtful reviews. We appreciate your positive feedback and would like to address your questions respectively:
>
> **Domain coverage:**
>
> As mentioned in the discussion section, there are many more domains and tasks that we could eventually incorporate into an expanded SEAM benchmark. We are planning to expand SEAM continuously, while we believe chess, chemistry, music, and graph already cover major domains like games, sciences, arts, and maths, respectively. We believe this set is a solid foundation on which the community can further build.
>
> **Task complexity control:**
>
> In principle, an intelligent vision-language reasoner, whether human or artificial, should exhibit consistent performance when presented with semantically equivalent information in different modalities. For example, a molecule shown as a structural diagram or described through a SMILES string contains the same information. A human expert chemist might prefer the visual structural diagram for intuitive reasoning, but when given a SMILES string, they can still identify the compound and its properties, possibly by first mentally or physically sketching the structure. The challenge lies in the molecular complexity and properties, not in the modality of its presentation. A truly intelligent VLM should behave analogously: the final answer should depend on the task itself, not on whether the representation is visual or textual. We thus follow this rationale to control the task complexity by ensuring semantically equivalent inputs in different modalities.
>
> **Improvement strategies to address modality gaps:**
>
> While our current work focuses on benchmarking modality imbalances of VLMs, your suggestions for improvement strategies are valuable future directions. Potential strategies could involve explicitly optimizing for cross-modal agreement during training or instruction-tuning to improve VLM performance, and using specialized tokenizers for domain-specific notations together with learning or finetuning special token embeddings. We will include more discussions in future manuscripts.
>
> **More comprehensive transformations as robustness testing:**
>
> While we already include a subset of possible image transformations (resolution, B&W, rotation), which we believe is sufficient to show the robustness, we agree that more comprehensive transformations would strengthen our analysis. Please advise what additional transformations are the most helpful ones to be added in your opinion.
>
> **Internal representation analysis:**
>
> Regarding where modality separation emerges, we conducted layer-wise embedding similarity analysis (shown in Figure 8). Our results show that modality differences begin immediately at the token embedding level (which we believe challenging to a belief that a fundamental capability of robust VLMs is to create unified representations across modalities right after the “merger”), improve somewhat in middle layers where cross-modal integration occurs, but then diverge again at output layers. This suggests that while models can partially bridge modality gaps during processing, the fundamental representational differences persist throughout the LLM decoder.

---

> > ### Comment · Reviewer_yu5A · 2025-06-03
> > **thank you for the response**
> >
> > Thank you for addresing my concerns. I will keep my score.

---

### Official Review · Reviewer_EVRg · 2025-05-12

**Rating:** 7
**Confidence:** 4
**Ethics Flag:** 1

**Summary:**

This work presents a new benchmark, SEAM, designed to test vision-language model capabilities on semantically equivalent tasks that are presented in different modalities. Notably, SEAM focuses on tasks where an instance can be independently represented in the vision and text modality with modality-specific notation, e.g. an image of a chess board and a text representation of the chess pieces/locations. They selected four tasks, including chess boards, chemical compounds, music compositions and graphs. In these tasks, every example includes text and an image that are semantically equivalent and used for a QA task.

Using SEAM, this paper evaluates different VLMs on the language only input, the vision only input and the vision+language input (assuming I’m correctly understanding Table 1 and Figure 3; I don’t see this explicitly described). Model performance tends differ between modalities across all models. Better performance in the visual domain may be due to poor language tokenization, such as for chemistry inputs, whereas better performance language performance may be due to weaknesses in visual perception. Overall, SEAM demonstrates that all models show a modality imbalance.

**Questions To Authors:**

*Questions*
- It’s not immediately clear to me what the “Vision-Language” results are in Table 1 and Figure 3. Is the model given both modalities as input for these experiments?

*Minor typos*
- L102: “ensure semantically equivalence” → “ensure semantic equivalence”
- L257: “a molecular shown” → “a molecule shown”

**Reasons To Accept:**

- SEAM allows for probing of the language capabilities and visual capabilities independently. This differs from most vision-language benchmarks where, even when seeking to probe each modality, the examples inherently require both inputs. I also like that every example is deterministically bidirectional.
- In addition to the benchmark, this paper also introduces a framework to continuously generate new examples. I appreciate this as we’re now in an era where evaluation datasets potentially leak into training data. Even though the tasks themselves are quite different, the authors chose easily available codebases for translating (text $\leftarrow$$\rightarrow$ vision) for each domain.

**Reasons To Reject:**

- The tasks here (chess, chemistry, music and graph) are fairly domain/knowledge specific. It’s tough to discern whether failures are due to poor (text-/visual-) reasoning or whether it’s due to a knowledge gap. For instance, the Music results in Figure 3 are fairly low across the board for every setup. One could ask – *is the modality gap here actually significant or is this just a challenging task?*
     - While this is touched on in L248-253, strong evidence isn’t really provided.

---

> ### Author Response · Authors · 2025-06-03
>
> We appreciate the positive feedback and would like to make a few clarifications.
>
> Firstly, the presence or absence of domain-specific knowledge does not inherently determine the modality gap we observe. The key insight of SEAM is that for semantically equivalent inputs, any knowledge requirements should be consistent across modalities. Specifically, if a model lacks chemical or musical knowledge, this knowledge deficiency should affect both text and visual processing equally by design. However, we consistently observe differences in performance across modalities even when the underlying semantic content and knowledge demands are identical.
>
> Furthermore, whether the tasks are challenging or not does not determine the modality gap directly, while we can only reveal the potential modality gap on moderately challenging tasks (otherwise, it might be all 0s or all 1s, thus the modality gap could not be found). That’s exactly why we need to control the task difficulty as introduced in Section 3.2.
>
> In terms of domain coverage, as mentioned in the discussion section, we realized more (in fact, innumerable) domains and tasks that we can incorporate into the SEAM benchmark. We are planning to expand SEAM continuously, while we believe chess, chemistry, music, and graph already cover major fields like games, sciences, arts, and maths, respectively.
>
> Minors:
>
> 1. “Vision-Language” denotes results that the models were given both modalities as input.
> 2. We will fix the typos in the revision. Thanks for pointing them out.

---

> > ### Comment · Reviewer_EVRg · 2025-06-05
> >
> > > Furthermore, whether the tasks are challenging or not does not determine the modality gap directly
> >
> > I agree about this point. My statement here is more minor -- for certain models in some domains, performance is low enough that the small modality gap might not provide signal. It's a minor weakness that doesn't impact the usefulness of SEAM, just something to note!
> >
> > That aside, I think this is a good paper that should be accepted. I'll keep my score.

---

### Official Review · Reviewer_hEvb · 2025-05-27

**Rating:** 6
**Confidence:** 3
**Ethics Flag:** 1

**Summary:**

The paper presents a new benchmark for measuring the gap in visual vs. language capabilities in vision language models across a number of different domains (chess, chemistry, graph theory and music). The data is collected or generated such that all information to answer the question is available in either visual or language form, which previous benchmarks did not guarantee. Moreover, the visual representation is not simply an image of a text transcription, like previous OCR-based benchmarks. The authors conduct an evaluation of existing models on the benchmark and show that there is indeed a gap between visual and language capabilities and present some early hypotheses as to why.

**Questions To Authors:**

See Reasons to Reject above for the critical issues. I think there are a number of empirical results which are needed to support the claims made in the paper.

On a minor note, I'd recommend passing the paper through a grammar aide and spell-checker. There are a large number of minor grammatical issues that can be easily fixed.

**Reasons To Accept:**

The problem seems well motivated. The benchmark design is done carefully and rigorously. From what I understand, it does nicely fit into the previous literature.

**Reasons To Reject:**

The empirical evaluation seems a bit thin after the presentation the benchmark. Table 1, Figure 2 and Figure 3 are all versions of the exact same data with different granularity. In addition, there are a number of claims in Section 4.2 that I don't think are well substantiated. I'll speak about them individually below:

> "One might hypothesize that as models become more accurate, they naturally converge... However, in the practical performance range of current models, evidence from Zhang et al. 2024b challenges this assumption... these results indicate that improving cross-modal alignment actively contributes to general reasoning capabilities"

I understand this claim as: the source of agreement cannot be attributed to accuracy improvements alone, where the support is drawn from a different paper entirely, not any additional experiments of this paper. I don't think Zhang et al. 2024b is great evidence for this particular claim because their evaluation used exactly the kind of text -> OCR benchmark that this paper avoids. I'd also want a specific pointer for which exact experiment of Zhang et al. 2024b the authors are referring to. Moreover, the last sentence in the quoted section above is a claim that is difficult to make precise. If it were true, one would expect that improving cross-modal alignment alone would result in, for example, stronger language reasoning abilities. But there is no such experiment in the text.

> "The severity of such textual perception limitation negatively correlates to the advantage of language inputs."

I believe the way this claim is written is a bit tautological. What I understand is that tokenization errors are negatively correlated to language-only performance. If this is the case, I would want to at least see a plot attempting to quantify this, but there are just a few qualitative case-studies in Appendix C.2 along with this paragraph. Additionally I would want at least some indication that this kind of issue is true for the vast majority of tokenizers for the models studied.

> "In domains where language inputs face known severe tokenization difficulties, such as chemistry and chess, the failure of vision inputs to compensate for these limitations is evident across most VLMs."

I interpret this claim as saying for cases where vision-alone is better than language-alone due to tokenization issues, vision-language fails to do better than vision-alone. This does not seem to be the case in Figure 2 (looking at Chemistry for Qwen2.5-VL-7B, InternVL-2.5-8B and Chess for Qwen2.5 72B, Qwen2.5 7B, InternVL 78B). These are the only cases where there's such a performance discrepancy. Many other models are still better using language-alone on Chess and Chemistry.

> "Such observations indicate that vision inputs are giving negative contributions, which empirically points to fundamental constraints in visual perception capabilities."

I am not sure that these "negative contributions" are in any way "fundamental." That would suggest that visual perception could not be improved, which is not a position I would take.

---

> ### Author Response · Authors · 2025-06-03
>
> We appreciate the recognition from the reviewer that “the problem seems well motivated” and “the benchmark design is done carefully and rigorously”.
>
> We understand that the main concern is that some claims made in follow-up discussions after presenting the benchmark and main performance comparisons are not well supported by sufficient experimental results. We sincerely appreciate the feedback from the reviewer, which helps us realize that some of our additional efforts, thoughts, and discussion beyond the scope of a benchmark paper will take much more future research work on top of such a benchmark to be well supported.
>
> We respectfully note that none of the pointed out claims is the core contribution of this benchmark paper, and some of those may require significant efforts to investigate. For example, validating “The severity of such textual perception limitation negatively correlates with the advantage of language inputs” requires a rigorous definition of the degree of tokenization error and a clear decomposition of perception and reasoning capabilities, which could potentially be an independent research work based on the proposed benchmark dataset.
>
> We would like to make adjustments according to the suggestions to moderate our claims and limit the scope of our work to proposing a useful benchmark (instead of expanding our work beyond a benchmark paper to support the claims).

---

> > ### Comment · Reviewer_hEvb · 2025-06-05
> >
> > Thank you for your response, I appreciate the discussion. I'll elaborate more below on my opinion:
> >
> > I would think the ideal benchmark paper does a few things:
> > 1. introduce the benchmark and detail the construction protocol
> > 2. show that there is some nontrivial signal for current systems to improve on
> > 3. point out some precise issues for further improvement that were difficult to measure previously
> >
> > I think that this paper does 1 and 2 effectively and my criticism in the review has mainly to do with 3. I find this response unsatisfying as the authors claim they'll walk back the strong claims in Section 4.2, which is the core discussion of the results, but without this Section we just have Table 1, and Figures 2, and 3 which are the same results in different formats (averaged or not).
> >
> > What this paper lacks then, in my opinion, is a case study like those attempted in Section 4.2, but which are currently lacking the empirical basis. If the proposed solution is to walk back the claims, I would like to understand what exactly will be the content of Section 4.2.

---

> > > ### Author Response · Authors · 2025-06-09
> > >
> > > Thank you for sharing your thoughts. We appreciate your recognition that 1 and 2 have been done effectively, and we agree with your suggestion to improve 3.
> > >
> > > We would like to address your concerns regarding 3 in two steps.
> > >
> > > First, please allow us to clarify some potential misunderstandings in the original review, as we believe these points could still be (partially) included in Section 4.2 to enhance 3.
> > >
> > > 1. Agreement and accuracy
> > >
> > >     As shown in Figure 2, we observe a strong positive correlation between accuracy and agreement. This correlation raises an important question about the **direction of causality**: (1) Does higher accuracy lead to higher agreement, or (2) does higher agreement drive better accuracy?
> > >
> > >     The purpose of discussing Zhang et al. 2024b is to support that (1) is not necessarily true, counter-intuitively. Specifically, such a finding was presented in Tables 1 and 4 in Zhang et al. 2024b (they use the word “consistency” to denote agreement). We respectfully disagree that whether the representations are OCR-based could affect this finding, as it is more about characterizing the relationship between the two **metrics** (agreement and accuracy) instead of task-specific performance. And in fact, their tasks are not limited to entirely OCR-based. For example, the State Machine Reasoning task shown in their Figure 3 uses visualized graphs and corresponding triples as vision and text inputs, respectively.
> > >
> > >     We originally thought that, since (1) could not be the only cause of the strong positive correlation between accuracy and agreement, (2) also contributes to such results. Therefore, we believe “improving cross-modal alignment actively contributes to general reasoning capabilities”. However, we realize from your review that this claim appears to be too strong, as we do not have experimental results where agreement is directly improved for comparisons. In fact, how to control the degree of cross-modal alignment remains an open research question.
> > >
> > >     In the revision, we would like to clearly explain the above reasoning, propose the potential benefits of improving agreement, and elaborate on the experimental settings for future works towards validating this claim.
> > >
> > > 2. “For cases where vision-alone is better than language-alone due to tokenization issues, vision-language fails to do better than vision-alone.”
> > >
> > >     This interpretation from the reviewer reflects exactly what we aimed to convey. However, we respectfully disagree with “This does not seem to be the case in Figure 2”, as this is the exact pattern found from our results in Figure 3. We did not aim to show this in Figure 2. Specifically, 13 settings satisfy the condition “vision-alone is better than language-alone”:
> > >
> > >     1. Chess: GPT-4o, GPT-4o-mini, Qwen-72B, Qwen-7B, Intern-78B, Llama-90B, Llama-11B, Gemma-27B, Gemma-12B
> > >     2. Chemistry: Qwen-7B, Intern-8B
> > >     3. Graph: Intern-8B, Llama-11B
> > >
> > >     And only 2 out of 13 settings (Chess Intern-78B and Graph Intern-8B) violate that “vision-language fails to do better than vision-alone” by a narrow margin.
> > >
> > >     Such results support our claim that “In domains where language inputs face known severe tokenization difficulties, such as chemistry and chess, the failure of vision inputs to compensate for these limitations is evident across most VLMs.”
> > >
> > > 3. Regarding the comment “Table 1, and Figures 2, and 3 which are the same results”, we agree that these results are computed from the same raw results, while they are shown in different granularities and forms to facilitate corresponding observations. Without Figure 2, it’s hard to see the relation between agreement and accuracy at a glance; without Figure 3, it’s hard to compare domain-specific modality imbalance. Additionally, Table 1 serves as a leaderboard with various aggregated metrics, e.g., Agreement-All, which are hard to incorporate into the other figures.
> > >
> > > We agree with the reviews that the two other pointed-out claims are not accurate enough, such as the mentioned “fundamental constraints” (too vague and bold) and “the severity of such textual perception limitation” (hard to measure the severity rigorously), which we will modify in the revision.

---

> > > > ### Author Response · Authors · 2025-06-09
> > > >
> > > > Second, following your suggestion regarding “case studies” and “empirical basis”. We would like to present a few more observations that indicate future improvement directions in VLMs.
> > > >
> > > > 1. To further investigate the tokenization error, we rerun the chess tasks with “gold” tokenization, which separates the tokens in FEN according to human commonsense. For example, if there are two Pawns next to each other, the FEN will include “PP” which should intuitively be tokenized into two “P” tokens. But VLMs often suboptimally regard “PP” as a single token.
> > > >
> > > >     We expect the models to perform better with the modified tokenizer. However, we found Qwen2.5-VL-72B-Instruct performs almost the same (original 0.542 vs gold 0.540). We hypothesize that two contradictory factors intervened to result in such performance: although the tokenization becomes intuitively better (positive factor), the models were not trained with such tokenization (negative factor). For example, “PP” might occur a lot in the training data, and the model always uses the single “PP” token to understand FEN. When we simply replace the “PP” token with two “P” tokens, the semantics learned in the embedding of “PP” could not be directly used, while the semantics of two adjacent “P”s were not as well trained.
> > > >
> > > >     Such observations suggest the importance of designing task-specific tokenizers and training domain-specific VLMs.
> > > >
> > > > 2. We also found error cases where vision preprocessing, in particular, the process of cutting each image into patches to feed ViTs, is problematic. Specifically, we found severe VLM hallucinations when vision inputs for Graph tasks are cut near the intersection of links. The hallucinations are particularly related to the edges and nodes involved in these intersections. Since we could not provide figures during rebuttal, we will directly add these cases to the revision.
> > > >
> > > >     Note that textual inputs of the Graph tasks are presented with adjacency matrices, in which numbers were well-separated by commas, and thus there are no known tokenization issues. Such observations could potentially explain why the vision performance is much worse than language in Figure 3 (Graph).

---

### Comment · Area_Chair_12vb · 2025-06-05
**Please review hEvb concerns!**

Dear all,

Thank you for taking the time to review this paper!

Please go through hEvb's concerns (thank you, hEvb, for listing them clearly!), consider them, and express your opinion.

I think this paper would benefit from a more thorough discussion before entering your final scores. This is the time to engage in one!

The discussion period is a critical part of the reviewing process, and ya'll are critical actors in its success.

Many thanks,
AC

---

### Decision · Program_Chairs · 2025-07-08

**Decision:**

Accept

**Comment:**

Four reviewers initially found the paper to be above or just around (above and below) the acceptance threshold.

Reviewers pointed out the following strengths:
  + “The problem seems well motivated. The benchmark design is done carefully and rigorously.”
  + “SEAM allows for probing of the language capabilities and visual capabilities independently. This differs from most vision-language benchmarks where, even when seeking to probe each modality, the examples inherently require both inputs.”
  + “Novel evaluation paradigm: SEAM provides a fundamentally new way to evaluate VLMs by testing their reasoning across distinct notation systems rather than just OCR-derived text, addressing a significant gap in existing benchmarks.”

In contrast, reviewers pointed out the following main weaknesses:
  - "The empirical evaluation seems a bit thin”
  - "a number of empirical results are needed to support the claims made in the paper.”
  - "Limited domain coverage"
  - "Limited investigation of improvement strategies"

After a lively discussion with the authors, many revisions were suggested and accepted by the authors that will make the presentation of the paper stronger. The main remaining questions revolve around whether this benchmark paper points out precise issues for further improvement that were difficult to measure previously, and whether the results are a bit too thin for a full paper.

After discussion, all reviewers recommend to accept the paper (scores: 6,6,7,6).

Having reviewed the paper, the reviews, and the discussion, the AC tends to agree that the paper is above the acceptance threshold and encourages the authors to incorporate the new insights from the discussion into their final manuscript.